# Factors Influencing Willingness to Undergo Lung Cancer Screening in the Future: A Cross-Sectional Study of Japanese University Students

**DOI:** 10.3390/healthcare12080849

**Published:** 2024-04-17

**Authors:** Yukihiro Mori, Manato Seguchi, Yoko Iio, Yuka Aoyama, Mamoru Tanaka, Hana Kozai, Morihiro Ito

**Affiliations:** 1Department of Nursing, College of Life and Health Science, Chubu University, 1200 Matsumoto-cho, Kasugai 487-8501, Aichi, Japan; moriyuki0821@isc.chubu.ac.jp; 2Graduate School of Life and Health Sciences, Chubu University, 1200 Matsumoto-cho, Kasugai 487-8501, Aichi, Japan; rb22004-1294@sti.chubu.ac.jp (M.S.); yoks@isc.chubu.ac.jp (Y.I.); aoyamay@isc.chubu.ac.jp (Y.A.); 3Department of Lifelong Sports and Health Sciences, College of Life and Health Sciences, Chubu University, 1200 Matsumoto-cho, Kasugai 487-8501, Aichi, Japan; 4Department of Clinical Engineering, College of Life and Health Sciences, Chubu University, 1200 Matsumoto-cho, Kasugai 487-8501, Aichi, Japan; 5Department of Food and Nutritional Sciences, College of Bioscience and Biotechnology, Chubu University, 1200 Matsumoto-cho, Kasugai 487-8501, Aichi, Japan; m-tanaka@isc.chubu.ac.jp (M.T.); hkozai@isc.chubu.ac.jp (H.K.)

**Keywords:** lung cancer, lung cancer screening, university students, smoking

## Abstract

Lung cancer (LC) is currently the leading cause of cancer deaths in Japan. Early detection through lung cancer screening (LCS) is important for reducing mortality. Therefore, exploring the factors affecting willingness to undergo LCS, particularly among young people, is important. This study aimed to elucidate the inclination toward LCS and its determining factors among Japanese university students. This cross-sectional study, involving 10,969 Japanese university students, was conducted in April 2023. A Pearson’s chi-square test and a binomial logistic regression analysis were used to analyze factors related to the dependent variable, willingness to undergo LCS in the future. Out of the 6779 participants (61.8%) involved in this study, 6504 (95.9%) provided valid responses, and 4609 (70.9%) expressed a willingness to undergo LCS in the future. Analysis revealed current smoking as a barrier to future willingness to undergo LCS. Other barriers included postponing the age of screening, anxiety about the screening content, and concerns about the possibility of having cancer after screening. Addressing barriers, such as current smoking and anxiety about screening, that prevent young people from undergoing LCS in the future is crucial. Therefore, universities should provide opportunities to educate students about LCS and explore various educational methods.

## 1. Introduction

Lung cancer (LC) is a common malignancy globally [1,2]. In Japan, LC represents the highest number of reported cancer deaths [3] and is a major clinical challenge. In general, early-stage LC manifests without symptoms and is often in an advanced stage by the time symptoms appear. Advanced LC is difficult to cure and involves substantial medical costs [4]. Therefore, early detection and treatment are essential. The early detection of LC through lung cancer screening (LCS) is crucial for reducing mortality [5]. Since 1987, Japan has implemented nationwide annual LCSs using chest radiography for people aged 40 years and above, supplementing it with sputum cytology for heavy smokers [5]. In Japan, low-dose computed tomography (LDCT) is also being studied in order to evaluate its effectiveness in screening high-risk individuals [6].

Smoking is widely recognized as one of the risk factors for LC [7,8]. Although it varies by country, the relative risk of LC for Japanese smokers compared to non-smokers is 4.4 times higher in men and 2.8 times higher in women [9]. Of further concern is that the median age at LC diagnosis is lower in populations with a higher smoking prevalence [10]. Despite this, a previous Japanese study reported that current smokers were less likely to undergo LCS compared to non-smokers [11]. Other barriers to receiving LCS include cognitive factors such as limited information [12] and ignorance [13], access-related factors such as lack of recommendations by health care providers [12], psychological factors such as concerns about LC [12] and negative attitudes towards LCS outcomes [13], and the inconvenience associated with accessing LCSs [13]. The LCS process varies across countries owing to varying evidence [14]. However, its effectiveness is often debated [15], with low screening rates observed among the target population [15] in countries where screening programs exist. Even in Japan, the LCS rate for those aged 40–79 is not high, at 53.4% for men and 45.6% for women [16]. Therefore, further research is needed to understand the barriers hindering a willingness to undergo LCS.

Therefore, this study aimed to determine the inclination of college students towards undergoing LCS and the factors influencing it. In this study, we hypothesized that in addition to smoking history [11], knowledge of LC and LCS, and the psychological factors [12,13] identified in previous studies, other factors, including experience with cancer education, could influence college students’ intentions to undergo LCS. Additionally, nicotine dependence [17] and the number of cigarettes smoked daily [11] are associated with smokers’ participation in several cancer screening programs. However, smoking paraphernalia has recently diversified in Japan [18]. Therefore, in this study, we also examined various smoking devices used by university students.

To our knowledge, no studies have investigated the factors influencing willingness to undergo LCS among college students. Our rationale for studying college students and focusing on LC is as follows: We considered college students to be a crucial demographic capable of enhancing their knowledge and awareness regarding LC prevention practices and smoking cessation. Although smoking is a major risk factor for LC, college students encompass different age groups that can legally start smoking. Furthermore, with the recent increase in the incidence of LC among young people [19], identifying factors that influence young people’s future intention to undergo LCS is important for preventive medicine and public health.

## 2. Materials and Methods

### 2.1. Study Design and Participants

This cross-sectional study was conducted in April 2023 at a single comprehensive university in Aichi, Japan. This university comprises many faculties, including humanities, social sciences, natural sciences, and medicine. We studied 10,969 undergraduate students enrolled at this university. Participants were recruited at the beginning of each semester during orientation meetings and other opportunities. Consequently, 6779 students (61.8%) participated in this study, of which 6504 (95.9%) provided valid responses.

A statistical power analysis, using the software IBM SPSS version 27 (IBM Corp., Armonk, NY, USA), was conducted in order to determine the sample size needed for this study. The results showed that a sample size of 183 was required to achieve a power of 0.8, a significance level of 0.05, and 28 explanatory variables. The sample size used in this study met this requirement.

This study was approved by the Chubu University Ethics Review Committee (Approval No.: 20220100).

### 2.2. Survey

Data collection was conducted using Google Forms (https://www.google.com/forms/about/). Participants responded to the survey anonymously.

#### 2.2.1. Willingness to Undergo LCS in the Future

Participants were asked to indicate their willingness to undergo LCS in the future by answering “yes” or “no”. There are two types of LCSs in Japan: population-based screenings, generally conducted by municipalities and other public entities, and opportunity-based screenings, which is voluntary for individuals. Both types are designed for asymptomatic individuals. In this study, LCS was defined as one of these two types, without limitation.

#### 2.2.2. Elementary Item

In addition to basic information such as grade, faculty, physical illness, hospitalization history, and cancer screening experience, participants also answered questions about their lifestyle, smoking habits, and familiarity with anti-smoking campaigns. The term “cancer screening experience” in this context referred to voluntary cancer screening, distinct from any general health checkups that participants receive at school or other institutions.

#### 2.2.3. Experience with Cancer Education, Willingness to Attend, Concerns about Cancer, and Knowledge and Awareness of LC and LCS

In order to ensure the validity of the questions, the present study was based on previous research [12,13], and questions regarding knowledge and perception of LC and LCS, psychological factors, and cancer education experience were formulated to examine their potential influence on college students’ intentions to receive LCSs.

Specifically, participants were asked about their past experiences with and receptiveness towards cancer education, alongside their concerns about cancer. Regarding general knowledge about LC and smoking, they were asked about the number of LC cases and deaths in Japan and the relationship between smoking (including passive smoking) and LC. Questions about LCS included the perceived health benefits of undergoing LCS and whether they were concerned about undergoing LCS due to lack of information about the LCS process. In addition, the reason for asking about age and LCS, regardless of their intention to undergo LCS, was to ascertain whether attitudes toward undergoing LCS at an earlier or later age influenced their willingness to undergo LCS.

#### 2.2.4. Additional Questions for Current Smokers

Smoking status, particularly number of cigarettes smoked per day [11], nicotine dependence [17], etc., has been reported to be associated with participation in several cancer screening programs, and several additional questions were provided for smokers.

Current smokers were asked about the smoking paraphernalia they use, the duration they have smoked, the number of cigarettes smoked daily, and their interest in smoking cessation. Additionally, the Fagerström Test for Nicotine Dependence (FTND) was used to measure physical nicotine dependence [20]. The FTND questions included (1) “How many cigarettes do you smoke daily?”, (2) “How soon do you smoke your first cigarette after waking up?”, (3) “Do you find it difficult to refrain from smoking in places where it is forbidden?”, (4) “Which cigarette would you hate most to give up?”, (5) “Do you smoke more frequently during the first hours after waking than during the rest of the day?”, and (6) “Do you smoke when you are seriously sick and in bed most of the day?” [21]. The FTND indicates low dependence with a total score of 0–3, moderate with 4–6, and high dependence with 7–10 [21]. In previous studies, Cronbach’s alpha coefficients for this scale ranged from 0.56 to 0.92, validating its reliability and generalizability [22]. Cronbach’s alpha coefficient in this study was 0.71, indicating relatively good reliability.

### 2.3. Statistical Analysis

A Pearson’s chi-square test was used to analyze differences in college students’ willingness to undergo LCS in the future, considering several explanatory variables, including smoking history and prior cancer education. Next, a binomial logistic regression analysis (increasing variable method) was conducted, with significant explanatory variables entered in order to identify important predictors of the dependent variable, willingness to undergo LCS. Furthermore, the same analysis was conducted in order to identify predictors of the intention to undergo LCS in the future among college students who are current smokers.

Dummy variables were assigned to the dependent variable “Are you willing to undergo LCS in the future?” and to the responses of each explanatory variable with “Yes” represented as 1 and “No” as 0. Adjusted odds ratios and 95% confidence intervals (95% CI) for each explanatory variable were calculated. Statistical significance was set at *p* < 0.05. SPSS version 27 was used for the statistical analyses.

## 3. Results

### 3.1. Study Participants

The characteristics of the participants are presented in Table 1. The mean age of the participants was 19.5 years, with a standard deviation of 1.3 years. Among the participants, 4290 (66.0%) were male, 1036 (15.9%) were undergraduate medical students, and 298 (4.6%) were current smokers.

### 3.2. Willingness to Undergo LCS and Related Factors

In total, 4609 respondents (70.9%) indicated their willingness to undergo LCS in the future. As shown in Table 2, differences in willingness to undergo LCS were observed based on sex (*p* < 0.01), faculty (*p* < 0.01), residential status (*p* < 0.05), hospitalization history (*p* < 0.05), and prior cancer screening (*p* < 0.01). Conversely, sleep duration, alcohol consumption, and exercise habits showed no significant differences. Analysis of smoking history revealed that the non-smoking group had a particularly high percentage of individuals willing to undergo LCS *(p* < 0.01). Furthermore, a higher percentage of those in the groups interested in social campaigns to promote smoking cessation and who were educated on smoking cessation were more willing to undergo LCS (all *p* < 0.01).

The results of the tabulation and analysis regarding willingness to undergo LCS (Table 3) encompass various factors such as experience of and education on cancer, concerns about cancer, knowledge about LC and smoking, and LCS awareness. These factors were significantly associated with a willingness to undergo LCS. For example, those educated on cancer were more likely to be willing to undergo LCS (*p* < 0.01). Other groups expressing concerns about potentially developing cancer in the future had a higher proportion of individuals willing to undergo LCS (*p* < 0.01). Conversely, individuals who expressed no concerns exhibited a greater willingness to undergo LCS compared to those who expressed concerns about the possibility of detecting LC after undergoing LCS (*p* < 0.01). Furthermore, when asked, “If you were to undergo LCS in the future, at what age would you be willing to have it?” the results revealed a higher percentage of respondents who indicated no intention of undergoing LCS in the “40s to 50s” and “60s or later” groups (*p* < 0.01).

The results of the analysis of participants who currently smoke are presented in Table 4 and Table 5. The group that had already undergone cancer screening had a higher percentage of those willing to undergo LCS (*p* < 0.01). Weak but significant differences were also observed for factors including faculty, drinking habits, interest in social campaigns promoting smoking cessation, education about smoking cessation, and intention to quit smoking in the future (all *p* < 0.05). Conversely, the statistical results showed no differences in smoking paraphernalia, smoking duration, number of cigarettes smoked daily, and nicotine dependence in terms of willingness to undergo LCS (Table 4). Except for a few factors, significant differences were observed in the association between factors such as experience with cancer education, willingness to undergo LCS, cancer anxiety, knowledge about LC and smoking, and LCS awareness (Table 5).

Table 6 presents the results from the logistic regression analysis. Across the entire sample, the following factors were positively correlated with a willingness to undergo LCS (all *p* < 0.01): interest in social campaigns promoting smoking cessation, having undergone cancer screening and education, willingness to receive cancer education in the future, harboring concerns about developing cancer in the future, awareness that smoking is a risk factor for developing LC, awareness of the existence of LCS, perceiving LCS as beneficial for one’s health, and knowing where to access LCSs. Conversely, being a current smoker was a negative factor (*p* < 0.05). Other negative factors were that if they were to undergo LCS in the future, they were in their “40s to 50s” (vs. 20s to 30s) and “60s or older” (vs. 20s to 30s), and being concerned about finding cancer after undergoing LCS (all *p* < 0.01).

In the analysis focused on current smokers, willingness to receive cancer education in the future, fear of developing cancer in the future, and feeling that undergoing LCS would be beneficial for their health were positive factors associated with a willingness to undergo LCS (all *p* < 0.01). Conversely, undergoing LCS at an age over 60 years (vs. 20–30 years) was considered a negative factor but was not statistically significant in this model (*p* = 0.062).

## 4. Discussion

This study investigated factors influencing willingness to undergo LCS in the future among Japanese university students. Current smoking was identified as a barrier influencing willingness to undergo LCS. Other barriers identified include postponing the age of screening, anxiety about the screening process, and fear of having cancer after screening. Among smokers, the smoking device used, smoking duration, number of cigarettes smoked daily, and nicotine dependence did not significantly influence intention to undergo LCS.

Several previous studies have reported an association between various cancer screenings and smoking status, indicating lower adherence to screening guidelines among smokers compared to non-smokers [23,24]. A previous study analyzing participants aged 40–64 years in Japan also reported that current smokers were less likely to undergo LCS compared to non-smokers [11]. The possibility that current college-age smokers may also be reluctant to undergo LCS in the future, as shown in this study, is noteworthy. Furthermore, prior studies have reported that among smokers, those with higher nicotine dependence were less likely to have undergone some form of cancer screenings [17]. The number of cigarettes smoked daily is also associated with participation in several cancer screenings, including LCS [11]. In contrast, our study showed that neither the number of cigarettes smoked daily nor nicotine dependence was associated with a willingness to undergo LCS among college students who currently smoke. Moreover, despite the rising popularity of new smoking devices, such as heat-not burn (HNB) cigarettes and electronic cigarettes, in the United States and Japan [25,26], the smoking device used might not influence intention to undergo LCS, as indicated in this study. Therefore, campaigns recommending LCS to college students who currently smoke should be implemented regardless of the number of cigarettes smoked, nicotine dependence level, or type of smoking equipment used.

Numerous studies have shown that the risk of various types of cancer increases with earlier initiation to smoking [27,28,29]. Additionally, quitting smoking at a younger age reduces the risk of developing cancer and death [30]. This study revealed that interest in social campaigns promoting smoking cessation and recognizing smoking as a risk factor for developing LC may positively influence the intention to undergo LCS. Furthermore, over 60% of the respondents in this study had received cancer education and were willing to be educated in the future. College student awareness of the risk of LC is associated with cancer prevention behaviors among college students, such as smoking cessation [31]. College students fall within the age group legally permitted to start smoking in Japan, underscoring the importance of efforts geared towards improving prevention strategies, such as smoking cessation programs for college students, along with recommendations for LCS visits.

In this study, anxiety about future cancer and the perceived benefits of undergoing LCS were considered positive factors influencing willingness to undergo LCS. Conversely, anxiety about the possibility of having cancer after undergoing LCS was a negative factor, and negative attitudes toward LCS outcomes may be a barrier to undergoing LCS at the individual level [13]. General benefits from LCS include a decreased likelihood of cancer-related mortality and the possibility of avoiding a lower quality of life and poor prognosis through early disease detection before progression. Providing education that helps participants overcome the dilemma of anxiety versus benefit is crucial.

Japanese cancer screening guidelines [32] recommend combining LCS with chest radiography and sputum cytology (currently for smokers) for both population-based and opportunistic screenings. Additionally, in opportunistic screening, LDCT is considered on an individual basis; the location of the LCS depends on these two approaches. Our findings indicate that anxiety regarding unfamiliarity with the LCS test procedure and knowledge about the LCS location influenced intention to undergo the LCS test. Overcoming anxiety about the checkup process and providing information about screening locations may also encourage future visitation.

LCS population-based screenings conducted by Japanese municipalities and other organizations typically recommend screening people over the age of 40 years annually. However, several studies have noted the increased incidence of LC in young people [19,33]. This study introduces a novel discovery that postponing consultation to a later age is associated with a reduced intention to seek medical attention. Since early detection and surgical treatment can potentially improve prognosis, early LCS should be actively promoted among young people [34].

A significant limitation of this study is the novelty of the data, as it represents the first insights from university students, rendering comparisons with prior studies challenging, despite approximately 70% of participants expressing a willingness to undergo LCS. This is because, to the best of our knowledge, this is the first data collected on college students. Other countries besides Japan that have LCS programs include Canada, South Korea, Croatia, and the United States [15]. Within the limited available literature, willingness to undergo LCS in the United States ranged from 82.8% to 92.8% [35,36]. Although it is difficult to generalize due to different age groups and attributes, it is plausible that, in comparison, the inclination of Japanese university students to undergo LCS is not as high. However, further discussion is warranted in order to determine whether this reflects differences in culture, values, and methods of public awareness across countries. In addition, the possibility of selection bias resulting from those who declined to participate in this study should be considered, particularly as this cross-sectional study was conducted at a single institution. Furthermore, although the university surveyed is coeducational, they unintentionally have a higher percentage of male students due to the composition of their faculties. Therefore, the results may not accurately reflect the general population. 

Although this study explored willingness to undergo LCS, the actual rate of LCS screening was not investigated. Regarding smoking, as of 2019, the adult smoking rate in Japan remains high at 27.1% for men and 7.6% for women [37]. However, the current smoking rate among the participants of this study was relatively low at 4.6%. In addition, approximately half of the smokers in this study used paper cigarettes, and approximately 17% used HNB cigarettes, deviating from the general usage rates in Japan [38,39]. Given these considerations, one must be cautious about generalizing the results. Furthermore, a high percentage of the participants in this study were “living with family”. This is expected to be due to the fact that many of the respondents are from the area where the target university is located, but we have not been able to ascertain the typical prevalence of smoking cessation and cancer prevention education in this area. Furthermore, this study investigated population-based and opportunistic LCS, including unstandardized tests such as chest radiography and LDCT. However, the process or accuracy of these screenings was not discussed. Finally, factors associated with the intention to undergo LCS other than those identified in this study may exist. For example, health beliefs and management status, as well as a history of cancer in close relatives. Further studies from a broader perspective are warranted.

However, the strength of this study is that it was conducted at a comprehensive university with faculties in several disciplines that attract students from across the country. To the best of our knowledge, this is the first study on willingness to undergo LCS among university students. The results of this study are important for preventive medicine and public health, and may be useful for developing educational programs on LC and LCS at universities.

## 5. Conclusions

This study assessed the willingness of 6504 Japanese university students to undergo LCS, as well as the barriers and facilitators associated with this willingness. Immediate action is required to address barriers, such as smoking and screening-related anxiety, identified in this study as factors that influence young people’s willingness to undergo LCS in the future. Therefore, we advocate for universities to offer comprehensive insights into LCS and smoking cessation, emphasizing the need to explore effective educational methods.

## Figures and Tables

**Table 1 healthcare-12-00849-t001:** Summary of participants’ characteristics.

		*n*	%
Age (Mean ± SD)	19.53 ± 1.30		
Sex			
Male		4290	66.0%
Female		2177	33.5%
Unanswered		37	0.6%
Education			
Freshman		2138	32.9%
Sophomore		1382	21.2%
Junior		1247	19.2%
Senior		1737	26.7%
Undergraduate department (course/program)			
Medical science		1036	15.9%
Others		5468	84.1%
Residence style			
Living with family		5331	82.0%
Living alone		1090	16.8%
Dormitory/others		83	1.2%
Has a physical illness and undergoing treatment.			
Yes		388	6.0%
No		6116	94.0%
Has a history of hospitalization.			
Yes		2066	31.8%
No		4438	68.2%
Lifestyle habits			
Sleep hours/day			
6 h or less/9 h or more		3892	59.8%
7–8 h		2612	40.2%
Drinking habits			
Non-drinker		4217	64.8%
Occasionally		2201	33.8%
Every day		86	1.3%
Exercise habits			
1 time or less/week		4177	64.2%
2 times or more/week		2327	35.8%
Smoking			
History of smoking			
Non-smokers		6062	93.2%
Past smokers		144	2.2%
Current smokers		298	4.6%
Family history of smoking			
Yes		3596	55.3%
No		2908	44.7%

SD, Standard deviation.

**Table 2 healthcare-12-00849-t002:** Relationship between participant characteristics and willingness to undergo lung cancer screening.

		All		Willingness to Undergo LCS				*p*-Value
				Yes (*n* = 4609, 70.9%)		No (*n* = 1895, 29.1%)		
		*n*	%	*n*	%	*n*	%	
Sex	Male	4290	66.0%	2967	64.4%	1323	69.8%	<0.01
	Female	2177	33.4%	1614	35.0%	563	29.7%	
	Unanswered	37	0.6%	28	0.6%	9	0.5%	
Education	Freshman	2138	32.9%	1548	33.6%	590	31.1%	ns
	Sophomore	1382	21.2%	975	21.2%	407	21.5%	
	Junior	1247	19.2%	883	19.2%	364	19.2%	
	Senior	1737	26.7%	1203	26.1%	534	28.2%	
Undergraduate department								
	Medical science	1036	15.9%	809	17.6%	227	12.0%	<0.01
Residence style	Living with family	5331	82.0%	3741	81.2%	1590	83.9%	<0.05
	Living alone	1090	16.8%	807	17.5%	283	14.9%	
	Dormitory/others	83	1.3%	61	1.3%	22	1.2%	
Has a physical illness and undergoing treatment								
	Yes	388	6.0%	276	6.0%	112	5.9%	ns
Has a history of hospitalization								
	Yes	2066	31.8%	1499	32.5%	567	29.9%	<0.05
Have had a cancer screening								
	Yes	439	6.7%	378	8.2%	61	3.2%	<0.01
Lifestyle habits								
Sleep hours/day	6 h or less 9 h or more ^†^	3892	59.8%	2727	59.2%	1165	61.5%	ns
Drinking habits	Non-drinker	4217	64.8%	3010	65.3%	1207	63.7%	ns
	Occasionally	2201	33.8%	1545	33.5%	656	34.6%	
	Every day	86	1.3%	54	1.2%	32	1.7%	
Exercise habits/week	1 time or less ^‡^	4177	64.2%	2965	64.3%	1212	64.0%	ns
Smoking related								
History of smoking	Non-smokers	6062	93.2%	4340	94.2%	1722	90.9%	<0.01
	Past smokers	144	2.2%	94	2.0%	50	2.6%	
	Current smokers	298	4.6%	175	3.8%	123	6.5%	
Family history of smoking								
	Yes	3596	55.3%	2568	55.7%	1028	54.2%	ns
Interested in social events promoting smoking cessation								
	Yes	3283	50.5%	2586	56.1%	697	36.8%	<0.01
Education on smoking cessation								
	Yes	4777	73.4%	3552	77.1%	1225	64.6%	<0.01

The *p*-values were calculated using a chi-square test. ^†^ The explanatory variable for comparison and contrast is ‘7–8 h’. ^‡^ The comparator is ‘2 times or more’. LCS, lung cancer screening; ns, no significant difference.

**Table 3 healthcare-12-00849-t003:** Relationship between answers to cancer-related questions and willingness to undergo lung cancer screening.

			All		Willingness to Undergo LCS				*p*-Value
					Yes (*n* = 4609, 70.9%)		No (*n* = 1895, 29.1%)		
			*n*	%	*n*	%	*n*	%	
Experience with cancer education, willingness to attend, and concerns about cancer									
Experienced cancer education									
		Yes	4237	65.1%	3223	69.9%	1014	53.5%	<0.01
Willing to receive cancer education in the future.									
		Yes	4500	69.2%	3703	80.3%	797	42.1%	<0.01
I am worried that I might get cancer in the future									
		Yes	4716	72.5%	3776	81.9%	940	49.6%	<0.01
Knowledge of LC and smoking									
I know that the number of patients with LC in Japan is increasing yearly									
		Yes	3581	55.1%	2599	56.4%	982	51.8%	<0.01
I know that in Japan, a high percentage of patients die from LC									
		Yes	4252	65.4%	3105	67.4%	1147	60.5%	<0.01
I know that there are few symptoms in the early stages of LC									
		Yes	3263	50.2%	2359	51.2%	904	47.7%	<0.05
I know that in Japan, LC ranks first among men and second among women in terms of deaths from different types of cancer									
		Yes	3044	46.8%	2201	47.8%	843	44.5%	<0.05
Smoking is recognized as a factor that contributes to LC development									
		Yes	5922	91.1%	4318	93.7%	1604	84.6%	<0.01
Acknowledges that “passive smoking” is a cause of LC									
		Yes	5965	91.7%	4334	94.0%	1631	86.1%	<0.01
Recognition of LCS									
Awareness of the existence of LCS.									
		Yes	2136	32.8%	1701	36.9%	435	23.0%	<0.01
If you are going to undergo LCS in the future, at what age would you be willing to have it?									
		20s to 30s	4602	70.8%	3516	76.3%	1086	57.3%	<0.01
		40s to 50s	1737	26.7%	1053	22.8%	684	36.1%	
		60s and beyond	165	2.5%	40	0.9%	125	6.6%	
Feeling that undergoing LCS is beneficial to their health									
		Yes	6231	95.8%	4558	98.9%	1673	88.3%	<0.01
Concerned about undergoing LCS because they lack information about its process									
		Yes	2448	37.6%	1619	35.1%	829	43.7%	<0.01
Concern that if they undergo LCS, they may find LC									
		Yes	1203	18.5%	770	16.7%	433	22.8%	<0.01
I know where to access LCS									
		Yes	1228	18.9%	774	16.8%	454	24.0%	<0.01

The *p*-values were calculated using the chi-square test. LC, lung cancer; LCS, lung cancer screening.

**Table 4 healthcare-12-00849-t004:** Relationship between characteristics of current smokers and their willingness to undergo lung cancer screening.

				Willingness to Undergo LCS				*p*-Value
				Yes (*n* = 175, 58.7%)		No (*n* = 123, 41.3%)		
		*n*	%	*n*	%	*n*	%	
Sex	Male	262	87.9%	159	90.9%	103	83.7%	ns
Education	Freshman	9	3.0%	4	2.3%	5	4.1%	ns
	Sophomore	18	6.0%	10	5.7%	8	6.5%	
	Junior	96	32.2%	57	32.6%	39	31.7%	
	Senior	175	58.7%	104	59.4%	71	57.7%	
Undergraduate Department								
	Medical science	40	13.4%	31	17.7%	9	7.3%	<0.05
Residence style	Living with family *	236	79.2%	136	77.7%	100	81.3%	ns
Has a physical illness and undergoing treatment								
	Yes	279	93.6%	164	93.7%	115	93.5%	ns
Has a history of hospitalization								
	Yes	213	71.5%	118	67.4%	95	77.2%	ns
Have had a cancer screening								
	Yes	48	15.1%	44	25.1%	4	3.3%	<0.01
Lifestyle habits								
Sleep hours/day	6 h or less 9 h or more ^†^	206	69.1%	121	69.1%	85	69.1%	ns
Drinking habits	Non-drinker	40	13.4%	20	11.4%	20	16.3%	<0.05
	Occasionally	236	79.2%	147	84.0%	89	72.4%	
	Every day	22	7.4%	8	4.6%	14	11.4%	
Exercise habits/week	1 time or less ^‡^	201	67.4%	116	66.3%	85	69.1%	ns
Smoking-related characteristics								
Smoking device	Cigarettes	148	49.7%	89	50.9%	59	48.0%	ns
	HNB cigarettes	53	17.8%	33	18.9%	20	16.3%	
	E-cigarettes	15	5.0%	7	4.0%	8	6.5%	
	Combined users	82	27.5%	46	26.3%	36	29.3%	
Smoking period	Less than 1 year	119	39.9%	65	37.1%	54	43.9%	ns
	2–3 years	163	54.7%	97	55.4%	66	53.7%	
	4 years or more	16	5.4%	13	7.4%	3	2.4%	
Number of cigarettes smoked/day	10 or less	208	69.8%	124	70.9%	84	68.3%	ns
	11–20	69	23.2%	38	21.7%	31	25.2%	
	21–30	13	4.4%	8	4.6%	5	4.1%	
	31 or more	8	2.7%	5	2.9%	3	2.4%	
Family history of smoking								
	Yes	246	82.6%	150	85.7%	96	78.0%	ns
Interested in social events promoting smoking cessation.								
	Yes	116	38.9%	78	44.6%	38	30.9%	<0.05
Attended educational programs on smoking cessation								
	Yes	184	61.7%	117	66.9%	67	54.5%	<0.05
Willingness to quit smoking in the future								
	Yes	98	32.9%	66	37.7%	32	26.0%	<0.05
FTND	Mild	174	58.4%	102	58.3%	72	58.5%	ns
	Moderate	99	33.2%	56	32.0%	43	35.0%	
	Severe	25	8.4%	17	9.7%	8	6.5%	

The *p*-values were calculated using the chi-square test. * The explanatory variable for comparison and contrast is ‘Living alone or with others’. ^†^ The comparator is ‘7–8 h’. ^‡^ The comparator is ‘2 times or more’. LCS, lung cancer screening; HNB, heat-not-burn; e–cigarettes, electronic cigarettes; combined users, combination of paper and HNB cigarettes or e-cigarettes; FTND, Fagerström Test for Nicotine Dependence; ns, no significant difference.

**Table 5 healthcare-12-00849-t005:** Relationship between answers to cancer-related questions and willingness to undergo lung cancer screening for current smokers.

		All		Willingness to Undergo LCS				*p*-Value
				Yes (*n* = 175, 58.7%)		No (*n* = 123, 41.3%)		
		*n*	%	*n*	%	*n*	%	
Experience with cancer education, willingness to attend, and concerns about cancer								
Experienced cancer education								
	Yes	191	64.1%	127	72.6%	64	52.0%	<0.01
Willing to receive cancer education in the future								
	Yes	152	51.0%	120	68.6%	32	26.0%	<0.01
I am worried that I might get cancer in the future								
	Yes	188	63.1%	145	82.9%	43	35.0%	<0.01
Knowledge of LC and smoking								
I know that the number of patients with LC in Japan is increasing yearly								
	Yes	213	71.5%	137	78.3%	76	61.8%	<0.01
I know that in Japan, a high percentage of patients die from LC								
	Yes	219	73.5%	136	77.7%	83	67.5%	<0.05
I know that there are few symptoms in the early stages of LC								
	Yes	185	62.1%	114	65.1%	71	57.7%	ns
I know that in Japan, LC ranks first among men and second among women in terms of deaths from different types of cancer								
	Yes	187	62.8%	116	66.3%	71	57.7%	ns
Smoking is recognized as a factor that contributes to LC development.								
	Yes	274	91.9%	168	96.0%	106	86.2%	<0.01
Acknowledges that “passive smoking” is a cause of LC								
	Yes	274	91.9%	169	96.6%	105	85.4%	<0.01
Recognition of LCS								
Awareness of the existence of LCS								
	Yes	139	46.6%	96	54.9%	43	35.0%	<0.01
If you are going to undergo LCS in the future, at what age would you be willing to have it?								
	20s to 30s	215	72.1%	142	81.1%	73	59.3%	<0.01
	40s to 50s	68	22.8%	32	18.3%	36	29.3%	
	60s and beyond	15	5.0%	1	0.6%	14	11.4%	
Feeling that undergoing LCS is beneficial to their health								
	Yes	264	88.6%	170	97.1%	94	76.4%	<0.01
Concerned about undergoing LCS because they lack information about its process.								
	Yes	100	33.6%	59	33.7%	41	33.3%	ns
Concern that if they undergo LCS, they may find LC								
	Yes	75	25.2%	43	24.6%	32	26.0%	ns
I know where to access LCS								
	Yes	224	75.2%	140	80.0%	84	68.3%	<0.05

The *p*-values were calculated using a chi-square test. LC, lung cancer; LCS, lung cancer screening; ns, no significant difference.

**Table 6 healthcare-12-00849-t006:** Factors influencing willingness to undergo lung cancer screening.

	OR	OR 95% CI		*p*-Value
		Lower Limit	Upper Limit	
All participants				
Current smoker. ^‡^	0.729	0.541	0.981	<0.05
Interested in social events promoting smoking cessation.	1.211	1.062	1.380	<0.01
Have had a cancer screening.	1.721	1.266	2.339	<0.01
Experienced cancer education.	1.268	1.143	1.487	<0.01
Willing to receive cancer education in the future.	3.494	3.060	3.990	<0.01
I am worried that I might get cancer in the future.	3.037	2.658	3.470	<0.01
Smoking is recognized as a factor influencing LC development.	1.419	1.144	1.761	<0.01
Awareness of the existence of LCS.	1.562	1.349	1.810	<0.01
If you are going to undergo LCS in the future, you should be in your 40s–50s. ^‡ ‡^	0.608	0.531	0.696	<0.01
If you are going to undergo LCS in the future, you should be in your 60s or older. ^‡ ‡^	0.197	0.129	0.300	<0.01
Feeling that undergoing LCS is beneficial to their health.	4.618	3.203	6.658	<0.01
Concerned about undergoing LCS because they lack information about the process.	0.608	0.527	0.701	<0.01
I am concerned that if I undergo LCS, they may find cancer.	0.774	0.651	0.921	<0.01
I know where to access LCS.	1.540	1.301	1.822	<0.01
Current smokers				
Willing to receive cancer education in the future.	3.485	1.949	6.230	<0.01
I am worried that I might get cancer in the future.	5.232	2.891	9.469	<0.01
If you are to undergo LCS, you must be in your 60s or older. ^‡ ‡^	0.116	0.012	1.116	0.062
Feeling that undergoing LCS is beneficial to their health.	4.979	1.614	15.362	<0.01

The *p*-values were calculated through binomial logistic regression analysis. The dependent variable was willingness to undergo lung cancer screening (Yes = 1, No = 0). All explanatory variables were answered with Yes/No (Yes = 1, No = 0) (references were all “No”). ^‡^, vs. non-smokers; ^‡ ‡^, vs. 20s to 30s; LC, lung cancer; LCS, lung cancer screening; OR, odds ratio; CI, confidence interval.

## Data Availability

The data presented in this study are available on request from the corresponding author due to privacy and ethical constraints.

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
