# Peer review of "Factors Influencing Willingness to Undergo Lung Cancer Screening in the Future: A Cross-Sectional Study of Japanese University Students"

_healthcare, 2024, doi:10.3390/healthcare12080849_

Round 1

Reviewer 1 Report

Comments and Suggestions for Authors

Dear Authors

The article by Mori et al, on factors influencing the willingness of Japanese University students to undergo lung cancer screening presents an interesting and important approach on lung cancer prevention strategies. 

It is noteworthy that 93.2% of the students are non-smokers. What is the epidemiology of smoking in Japan, since "LC ranks first among men and second among women"? 

Speaking if women, as already mentioned in the limitations, I was wondering of the sex bias: is it because in Japan there are less female students going to the University, or is there another reason why they were not included int he study? ie 4290 vs 2177 is a huge difference in times of sex equity.

Concerning the background of the students, do they all come from the area where the University is located, since most of them n=5331 "live with family". Is this an urban or a rural area? Is it an underserved area where smoking prevention education was difficult to promote or never promoted?

Only 4.6% are current smokers which is very encouraging especially, since your results suggest that there is a gap in lung cancer education and knowledge.

I would kindly suggest to clarify or briefly add / discuss the aforementioned points to make this research of high impact potential in LC screening prevention strategy.

Author Response

Dear reviewer

Thank you very much for reviewing our paper.
We would like to respond to your comments.

Reviewer 2 Report

Comments and Suggestions for Authors

Dear Authors and Editors,

The paper "Factors Influencing the Willingness to Undergo Lung Cancer Screening in the Future: A Cross-Sectional Study of Japanese University Students" by Morihiro Ito et al. addresses a very relevant question about attitudes and factors influencing willingness to undergo lung cancer screening.

The introduction provides a very good introduction to the question with the corresponding data and publication situation. The relevant studies seem to be presented. The methods section is relatively short and the content of the methodological preparation of the questionnaire could be supplemented. Furthermore, it remains unclear whether there are attention control questions or control questions for internal validity. This should definitely be added and in this context it would be suggested to subdivide the methods section with subheadings.

The presentation of the results and the statistics used seem plausible and purposeful. The discussion takes up the question well and deals with the discussion and classification of the results in the existing literature on the basis of the results obtained. The topic of limitations is somewhat underrepresented or not clearly articulated and should be addressed more clearly at this point.

In summary, I would support publication once the points of criticism have been addressed.

Author Response

(The authors gave the same response as above.)

Round 2

Reviewer 2 Report

Comments and Suggestions for Authors

Dear editor, dear authors,

thank you very much for the revision, through which the work, in my view, has gained a lot, so that I would speak out in favor of acceptance and publication.

Author Response

Dear reviewer.

Thank you for reviewing my paper.

We thank you from the bottom of our hearts.

Sincerely,

Morihiro Ito